# Forecasting extreme stratospheric polar vortex events

L. J. Gray [1,2 ✉], M. J. Brown [1], J. Knight[3], M. Andrews[3], H. Lu [4], C. O'Reilly [1,2] & J. Anstey [5]

Extreme polar vortex events known as sudden stratospheric warmings can influence surface winter weather conditions, but their timing is difficult to predict. Here, we examine factors that influence their occurrence, with a focus on their timing and vertical extent. We consider the roles of the troposphere and equatorial stratosphere separately, using a split vortex event in January 2009 as the primary case study. This event cannot be reproduced by constraining wind and temperatures in the troposphere alone, even when the equatorial lower stratosphere is in the correct phase of the quasi biennial oscillation. When the flow in the equatorial upper stratosphere is also constrained, the timing and spatial evolution of the vortex event is captured remarkably well. This highlights an influence from this region previously unrecognised by the seasonal forecast community. We suggest that better representation of the flow in this region is likely to improve predictability of extreme polar vortex events and hence their associated impacts at the surface.

---

[1] National Centre for Atmospheric Science, Oxford, OX1 3PU, UK. [2] Department of Physics, Oxford University, Oxford, OX1 3PU, UK. [3] Met Office Hadley Centre, Exeter, UK. [4] British Antarctic Survey, Cambridge, UK. [5] Canadian Centre for Climate Modelling and Analysis, Victoria, Canada. ✉email: lesley.gray@physics.ox.ac.uk

Seasonal weather forecast models aim to capture sources of atmospheric predictability so that their impact on surface weather can be exploited. These are usually associated with the long-term memory of the oceans, sea ice or snow cover, but the role of the lower stratosphere has also been recognised[1–6].

The most extreme event in the winter stratosphere is the major sudden stratospheric warming (SSW)[7]. Polar temperatures increase by tens of degrees in a few days, and the normal westerly stratospheric polar vortex reverses to easterly. This abrupt change in circulation can impact the underlying tropospheric flow for 2 months or more[3], including the North Atlantic Oscillation (NAO), an index connected to the North Atlantic jet-stream and European weather[1–6,8–11]. Following a major warming, the NAO is more likely to be in its negative phase, resulting in increased chances of wet and windy weather over Southern Europe and extremely cold temperatures over Northern Europe[12,13]. Winter NAO forecast skill is substantially reduced when winters with warming events are excluded[6,14].

The full impact of major warmings on seasonal forecasts can only be achieved if they themselves can be forecast well in advance. Major warmings are sporadic events. Some winters have one or occasionally two, while others have none. Influencing factors can be related either to (a) tropospheric wave forcing, or (b) the ambient background stratospheric winds through which the waves propagate[15,16]. For example, major warmings are influenced by El Nino events, the Madden–Julian Oscillation (MJO) and blocking events, all of which act via (a), and the phase of the quasi-biennial oscillation (QBO), which acts via (b)[17–24]. However, while these factors may provide a statistical indication of the likelihood of a warming event (SSW) sometime during the winter, the actual timing and whether it develops from a minor to major SSW is much more difficult to predict. Even when known influential mechanisms are represented in state-of-the-art seasonal forecast models, the current consensus is that warmings are generally predictable with high confidence only ~10–15 days ahead[4,25–27], albeit with some exceptions[28]. Probabilistic indications of the timing of events with lower levels of confidence are often available beyond this timescale, but with decreasing skill at progressively longer lead time[29].

In this study, we investigate factors that influence the timing and vertical extent of major SSWs through ensemble model experiments, using the major split vortex event in January 2009 as our primary case study (Fig. 1a; see also Supplementary Note 1 and Supplementary Movie 1). Our aim is to explore the potential for improved probabilistic seasonal weather forecasts through improved forecasts of major warmings. The experiments highlight sensitivity of SSWs to the evolution of flow in the equatorial upper atmosphere, the importance of which has hitherto been unrecognised by the seasonal forecasting community. While the SSW event cannot be reproduced by constraining winds and temperatures in the troposphere alone, an additional constraint applied only to the zonal winds in the equatorial upper atmosphere achieves a remarkably good simulation with little ensemble spread. The equatorial upper atmosphere is dominated by the semi-annual oscillation (SAO). Our results suggest that improved representation of the SAO to correct an easterly bias found in most forecast models is likely to improve probabilistic predictions of the timing and depth of SSW events and hence their impacts on surface weather.

## Results

**The control experiment.** A 50-member ensemble control experiment of the atmospheric model was performed, initialised to the observed September 2008 winds and temperatures (see "Methods") and then allowed to freely evolve through winter with

no further observational constraints apart from the imposed sea surface temperatures (SSTs). The model adequately reproduces the climatological evolution of the stratospheric vortex winds (Fig. 1b), with the ensemble-mean westerlies gradually building from September to reach maximum strength in December–January and thereafter weakening and reversing to summer easterlies near the end of April. The sporadic weakening of the winds occurs during the winter in response to wave forcing from below, but a clear SSW event (indicated by a reversal of the ensemble-mean winds to easterlies in mid-winter) is not evident. The ensemble shows large spread (Supplementary Fig. 1), with 22 of the 50 members displaying a warming event at sometime during December–January–February, but only two of these occurs within 15 days of the observed event (Supplementary Table 1).

**The role of tropospheric wave forcing.** We first explore how well a correct representation of the troposphere might constrain the stratospheric flow. In the AllTrop experiment, the zonal wind (**u**), meridional wind (**v**) and temperature (T) fields were relaxed towards ERA-Interim fields at all latitudes from just above the surface to the tropopause (~300 hPa), thus ensuring the accuracy of the extratropical wave forcing from the troposphere. The ensemble-mean vortex wind evolution now shows evidence of two SSW events (Fig. 1c) and the ensemble spread is substantially reduced (Fig. 2), but the timing and characteristics are inaccurate. The vortex strengthens from October, but then weakens significantly in early November and again in early December when the winds reverse at the upper levels, producing a so-called minor SSW (the wind reversal does not quite extend down to 10 hPa, as required to qualify as major warming). These events roughly coincide with features in the observations; for example, in early December, there is evidence of a zero contour at ~0.5 hPa in Fig. 1a, but the AllTrop feature is much more pronounced. Thereafter, the winter evolution diverges further from the observed behaviour, with a slightly weakened vortex in late January but no indication of a major SSW until the end of February, a month later than observed.

In summary, while imposing the tropospheric fields improves the simulation, the experiment confirms that this is insufficient on its own to accurately simulate the SSW[30]. Correlation of the ensemble-mean 10 hPa, 60°N zonal-averaged winds with the corresponding ERA-Interim data substantially improves (0.56; see Supplementary Table 1) compared with the control experiment (0.29), but it fails to reproduce the timing and depth of the observed SSW.

**The role of the stratospheric background flow.** A corresponding set of experiments was performed to test the role of the background stratospheric flow. In the UpStrat experiments (Fig. 1d), the relaxation towards ERA-Interim **u**, **v** and T fields was applied only in the upper stratosphere (above 5 hPa at all latitudes). Not surprisingly, a good representation of the timing of the SSW is achieved since the relaxation is applied directly to the upper part of the vortex. However, the SSW does not penetrate sufficiently deep into the lower stratosphere. While some of the ensemble members develop easterlies at 10 hPa (Supplementary Fig. 1), there is significant ensemble spread, and the disparity with the observed evolution increases further at the lower levels.

Prompted by earlier studies[31–33], a further experiment was performed to test whether there is a remote influence on the timing of the SSW from the equatorial upper atmosphere. The AllTrop-UpStrat-Eq experiment had a combination of relaxation up to the tropopause (as in AllTrop) and in the stratosphere above 5 hPa, but the stratospheric relaxation was applied only

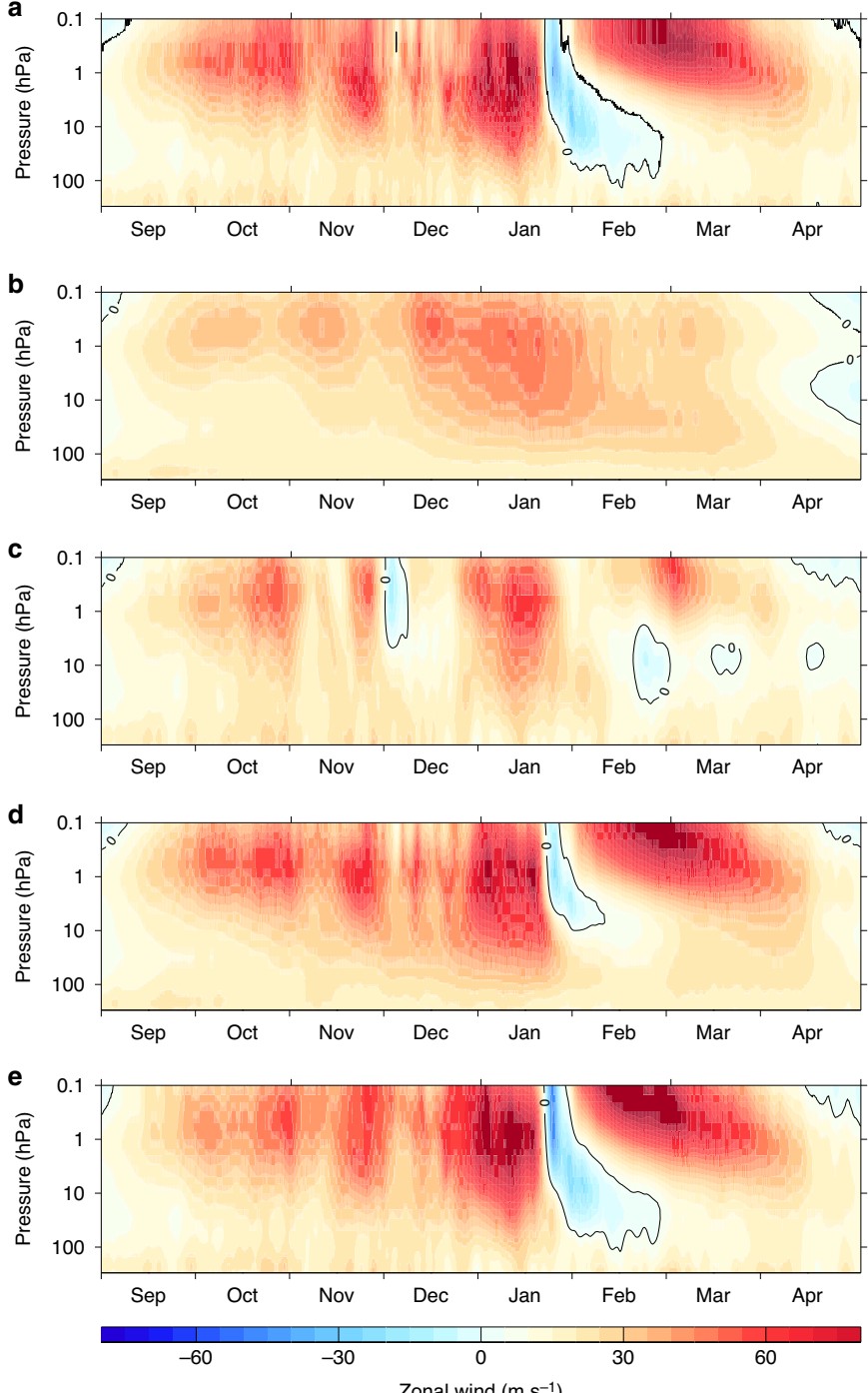

**Fig. 1 Polar vortex evolution under different relaxation scenarios.** Comparison of zonally averaged zonal wind (ms$^{-1}$) evolution at 60ºN for 2008/9 from the different model experiments. **a** Evolution of the European Centre Interim Reanalysis (ERA-Interim) data to indicate the observed evolution. **b** Ensemble average of the control run in which no relaxation was applied to the model; with only imposed sea surface temperatures at the lower surface and the correct phase of the quasi-biennial oscillation (QBO) from the initial conditions the model achieves a reasonable simulation of the vortex seasonal evolution but is unable to reproduce the observed sudden warming event in January 2009. **c** Ensemble average of the AllTrop experiment in which winds and temperatures from the surface to the tropopause at all latitudes were relaxed towards the ERA-Interim data; this demonstrates that imposing the tropospheric wave forcing enables the model to successfully simulate a sudden warming event, but its timing and penetration depth are incorrect. **d** Ensemble average of the UpStrat experiment in which the winds and temperatures in the upper stratosphere above 5 hPa at all latitudes were relaxed towards ERA-Interim data; this indicates that even when the correct timing of the warming event is imposed the model is unable to correctly reproduce the vertical extent of the warming event deep into the lower stratosphere. **e** Ensemble average of the AllTrop-UpStrat-Eq experiment in which the winds and temperatures from the surface to the tropopause were relaxed toward the ERA-Interim data at all latitudes (as in the AllTrop experiment), and additionally the zonal winds in the upper equatorial stratosphere between 0 and 10ºN above 5 hPa were relaxed towards ERA-Interim data; the polar vortex evolution from this experiment is remarkably similar to the observed evolution, indicating that the combination of realistic tropospheric wave forcing and realistic equatorial winds at all heights in the stratosphere and lower mesosphere (i.e., the correct phasing of the quasi-biennial oscillation in the lower stratosphere from the initial conditions and the semi-annual oscillation (SAO) from the imposed relaxation) is required to successfully reproduce the timing and depth of the observed warming event.

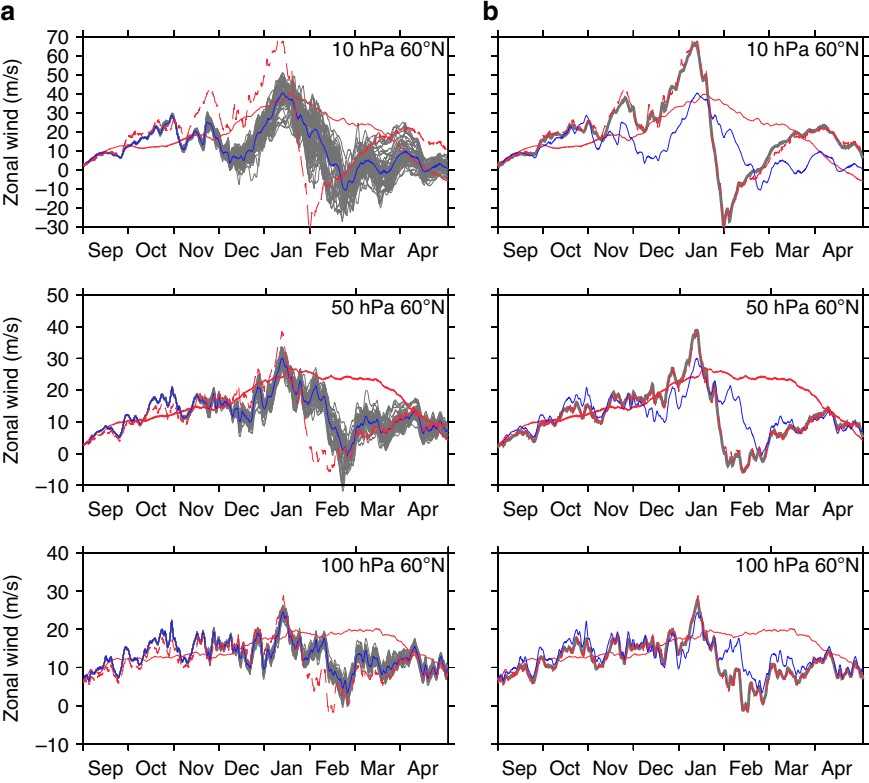

**Fig. 2 Ensemble spread of the polar vortex evolution.** Evolution of zonally averaged zonal winds ($ms^{-1}$) at 60ºN for all ensemble members at selected pressure levels from (**a**) the AllTrop experiment in which winds and temperatures from the surface to the tropopause at all latitudes were relaxed towards the European Centre Reanalysis Interim (ERA-Interim) data and **b** the AllTrop-UpStrat-Eq experiment in which the winds and temperatures from the surface to the tropopause were relaxed towards ERA-Interim data at all latitudes (as in the AllTrop experiment) and additionally the zonal winds in the upper equatorial stratosphere between 0 and 10ºN above 5 hPa were relaxed towards ERA-Interim data. Red dashed lines show the ERA-Interim data, and red solid lines show the evolution of the control run in which no relaxation was applied. Grey lines show individual ensemble members with a thick black line showing the ensemble mean. The AllTrop ensemble-mean is shown in blue in all plots for comparison. Note that the ensemble-mean of the AllTrop-UpStrat-Eq experiment that successfully reproduced the timing and vertical extent of the observed warming event is barely distinguishable from the observations (red dashed) and the individual ensemble members (grey) are barely evident because the ensemble spread is so small.

between 0 and 10ºN and only to the zonal wind field **u** (so that the meridional winds and temperatures are free to adjust accordingly). The results of the AllTrop-UpStrat-Eq experiment are quite remarkable. The timing (Fig. 1e) and split vortex evolution (Supplementary Movie 2) are captured well, correlation of the ensemble-mean with the ERA-Interim zonal winds at 60ºN, 10 hPa reaches 0.98 (Supplementary Table 1), the timing of all 50 ensemble members is accurate, and there is so little spread they are virtually indistinguishable (Fig. 2).

In addition to the 2008/9 case study, corresponding experiments (control, AllTrop and AllTrop-UpStrat-Eq) were repeated for two more SSW events, in February 1989 (Supplementary Fig. 2) and January 2006 (Supplementary Fig. 3). The results suggest that the success of AllTrop-UpStrat-Eq in 2008/9 was not fortuitous (Supplementary Table 1). A remarkably successful prediction of the timing and penetration depth was achieved in each of these winters, with little ensemble spread.

**Wave forcing of the mean flow.** The mechanism of influence from the equatorial upper stratosphere/mesosphere in AllTrop-UpStrat-Eq can be explored using Eliassen–Palm (E–P) flux diagnostics to illustrate wave mean-flow interaction. We examine these at various stages in the winter evolution (Fig. 3). The two E–P flux components (arrows) indicate characteristics of the wave propagation, and the background contours show the zonally

averaged zonal winds through which the waves propagate. Distributions of the E–P flux divergence (Supplementary Fig. 4) indicate wave forcing of the mean flow. We compare with corresponding diagnostics from AllTrop that failed to achieve realistic timing of the warming, noting that the only difference is the relaxation of the zonal winds at 0–10ºN above 5 hPa.

Figure 3 indicates that as winter progresses, Rossby waves increasingly propagate upward from the troposphere at mid-latitudes. As their amplitude increases with decreasing density, they preferentially break in the mid-to-upper stratosphere, where their easterly momentum is transferred to the background flow. However, their propagation is influenced or impeded by the presence of easterly or, conversely, very strong westerly background flow[34]. The edge of the polar vortex, where potential vorticity gradients are large, acts as a waveguide. In the lower to the middle stratosphere where the background zonal winds are within this range, the waves tend to propagate along the polar vortex edge. In the upper stratosphere where the westerlies can reach $>50\,m\,s^{-1}$, the waves tend to propagate along the equatorward or poleward flank of the strong vortex westerlies, depending on the latitude of the vortex relative to the latitude of the waves[35,36].

In October, there are relatively small differences between the wind and E–P flux distributions in the two experiments. In November (and December), downward pointing E–P flux arrows at mid and high latitudes in the difference plots indicate reduced

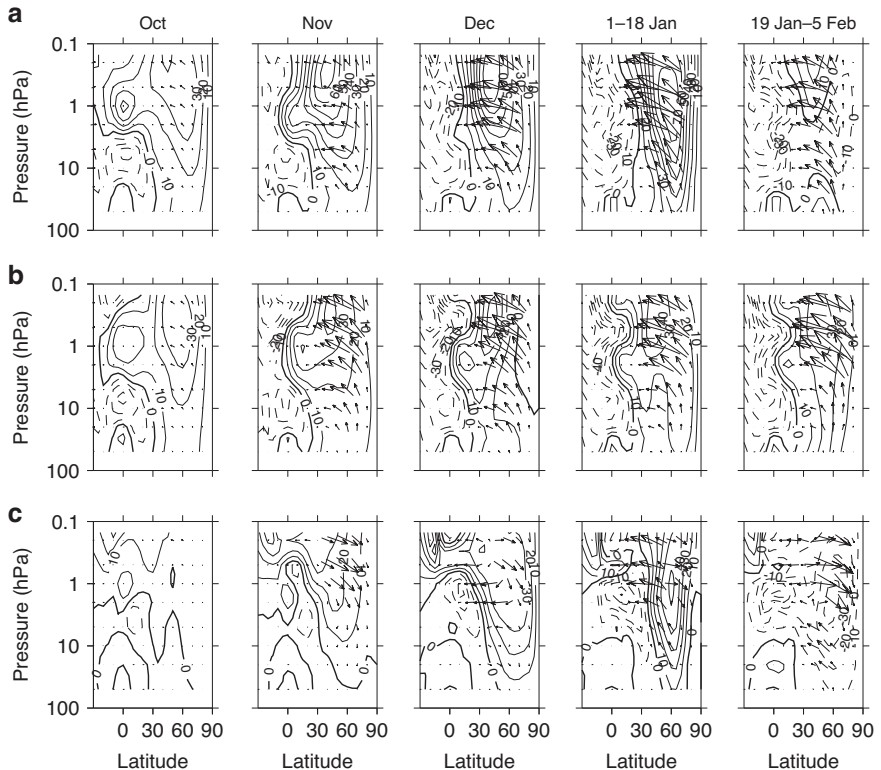

**Fig. 3 Background wind and wave propagation diagnostics.** Pressure (hPa) versus latitude plots (30ºS to 90ºN) of the simulated 2008/9 Eliassen–Palm (E–P) flux arrows (m² s⁻²) to indicate wave propagation. Contours show the zonally averaged zonal winds (ms⁻¹) with a contour interval of 10 ms⁻¹. Dashed contours denote negative (easterly) winds. The combination of E–P fluxes and background zonal winds allows an examination of the wave mean-flow evolution and interaction during the winter (see Supplementary Fig. 4 for the corresponding distributions of E–P flux divergence that provide an indication of the wave mean-flow momentum transfer). Monthly averaged fields are shown for October–December leading up to the sudden warming, while the fields averaged over the periods 1–18 January and 19 January–5 February show distributions immediately before and after the SSW event occurred. **a** shows the AllTrop-UpStrat-Eq experiment that was successful in reproducing the observed SSW event, in which the winds and temperatures from the surface to the tropopause were relaxed towards the European Centre Reanalysis (ERA-Interim) data at all latitudes and additionally the zonal winds in the upper equatorial stratosphere between 0 and 10ºN above 5 hPa were relaxed towards ERA-Interim data. Note that the AllTrop-UpStrat-Eq equatorial zonal winds above 5 hPa are identical to ERA-Interim, by design. **b** shows the AllTrop experiment that failed to correctly reproduce the timing of the SSW event, in which only the winds and temperatures from the surface to the tropopause were relaxed towards the ERA-Interim data at all latitudes. **c** shows the difference between the two experiments (AllTrop-UpStrat-Eq minus AllTrop).

wave forcing of the background flow in AllTrop-UpStrat-Eq even though the tropospheric wave forcing is identical in both experiments (by design). The equatorward E–P flux arrows in AllTrop increase in magnitude in the upper stratosphere and mesosphere, indicating that the waves propagate into the mesosphere where they break. The resulting wave mean-flow interaction at 0.1–1 hPa weakens the upper-level westerlies, allowing persistent upward wave penetration and wave absorption. This strong wave absorption is consistent with the weaker polar vortex throughout the whole of its depth in AllTrop.

In contrast, the AllTrop-UpStrat-Eq EP flux arrows are noticeably smaller above 1 hPa but become near-horizontal below ~1 hPa, indicating a lack of penetration of the waves into the upper stratosphere/mesosphere. The lack of wave forcing above 1 hPa means that the upper-level westerlies continue to strengthen, further inhibiting subsequent waves into the region. As a result of this reduced early winter wave mean-flow interaction, the AllTrop-UpStrat-Eq vortex remains relatively undisturbed at the upper levels through December and early January. At the same time, enhanced wave absorption below 1 hPa in the subtropics leads to a poleward shift of the equatorial waveguide to higher latitudes in the middle to lower stratosphere. This latter process is highlighted by the region of easterly wind differences established in the subtropics in November/December (Fig. 3,

0–20ºN 1–10 hPa, bottom row, see also Supplementary Fig. 4) that grows and expands poleward through the winter, eventually leading to the SSW in late January.

**The role of the equatorial upper stratosphere.** The imposed relaxation in the equatorial upper atmosphere in AllTrop-UpStrat-Eq thus appears to provide the correct boundary conditions for realistic development of the split SSW in late January, by defining the cavity for Rossby waves where they interact with the mean flow (possibly via a self-tuning mechanism moving towards its resonant point[37]). Relatively small differences in the equatorial/subtropical upper atmosphere appear to strongly influence the timing of the SSWs, and all ensemble members achieve the warming event at virtually the same time.

The equatorial upper stratosphere/mesosphere is dominated by the SAO, with regular 6-monthly alternating easterly/westerly winds at solstice/equinox[38] (Fig. 4). Without the relaxation in the equatorial upper stratosphere, the model develops a clear easterly bias in early winter compared with ERA-Interim fields. The peak of the SAO westerlies in October above 5 hPa is stronger in AllTrop-UpStrat-Eq (~40 ms⁻¹) than in AllTrop (~25 ms⁻¹) and the westerlies extend upwards into the lower mesosphere, while in AllTrop there is a reversal to easterlies at ~0.2 hPa (note that the evolution in AllTrop-UpStrat-Eq above ~5 hPa is identical to the

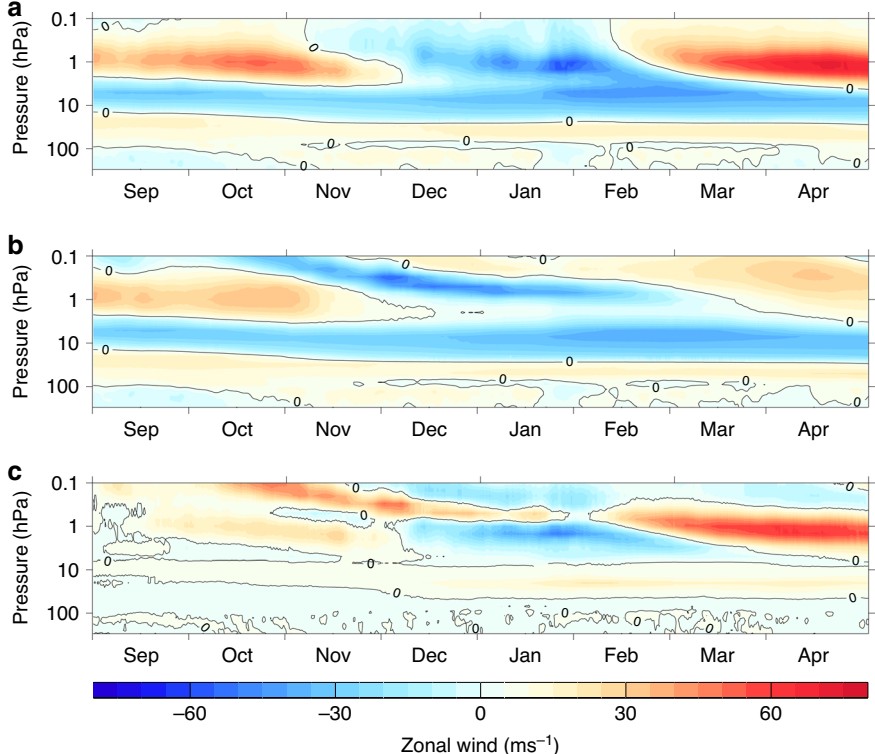

**Fig. 4 Evolution of the semi-annual oscillation (SAO).** Comparison of the time evolution of 2008/9 zonally averaged zonal winds (ms$^{-1}$) at equatorial latitudes. **a** shows the AllTrop-UpStrat-Eq experiment that successfully reproduced the timing of the observed January 2009 SSW event. **b** shows the AllTrop experiment that achieved an SSW event but with incorrect timing. **c** shows the difference between the two experiments (AllTrop-UpStrat-Eq minus AllTrop). Both experiments were initialised in September with the correct phase of the quasi-biennial oscillation in the lower stratosphere (i.e., the westerly winds at 20–100 hPa in 4a and 4b), and this is successfully maintained in both experiments by the gravity wave scheme. The only difference in the experimental set-up is that the AllTrop-UpStrat-Eq experiment included an additional relaxation of the zonal winds towards the European Centre Reanalysis (ERA-Interim) data in the upper equatorial stratosphere (between 0 and 10ºN above 5 hPa) to ensure the correct evolution of the SAO. The equatorial winds above 5 hPa in the AllTrop-UpStrat-Eq experiment are therefore identical to the ERA-Interim data whereas they evolve freely in the AllTrop experiment. The difference in the two evolutions indicates an underlying easterly bias in the early winter of the freely evolving model.

ERA-Interim fields because of the applied relaxation). Comparisons with other reanalyses[39] and with direct satellite observations[39,40] indicate that this westerly SAO phase extension into the mesosphere in ERA-Interim is realistic. Indeed, Microwave Limb Sounder (MLS) and Sounding of the Atmosphere using Broadband Emission Radiometry (SABER) satellite observations suggest that ERA-Interim may underestimate their full strength[40,41].

An easterly bias in the upper stratosphere/mesosphere is common to most models[40] and is attributed to a deficit of westerly wave forcing associated with absorption of vertically propagating equatorial Kelvin and gravity waves. Modelling this wave forcing is challenging. It requires parametrisation of non-orographic gravity waves (see "Methods"), but the wave source amplitudes are not well known, and their propagation and absorption depends on accurate representation of the background flow. In addition, wave amplitudes in this region are large, and the flow is highly non-linear. The wave structures themselves, including the longitudinal structure of the cross-equatorial flow may, therefore, be important[42,43].

To investigate the influence of the SAO alone (i.e., excluding the AllTrop part of the relaxation), two experiments were performed, identical to the control experiment apart from relaxation of the equatorial zonal winds between 0 and 10ºN above 5 hPa. The UpStrat-Eq-ERA (Supplementary Fig. 6) and UpStrat-Eq-MERRA (Supplementary Fig. 7) experiments were relaxed towards the 2008/9 ERA-Interim and MERRA2 zonal

winds, respectively. The two sets of experiments using different reanalyses were performed because there are relatively large differences between the reanalyses in this region[39].

As with the control experiment, there is a large ensemble spread because the tropospheric wave forcing is relatively unconstrained with only imposed SSTs. The ensemble-mean winds in both sets of experiments remain too weak in the period leading up to the observed SSW event in January; this cannot be rectified simply by correcting the SAO region. (As an aside, this suggests an underlying positive bias in tropospheric wave forcing in the unconstrained model; in all three case studies the weak vortex bias is removed in the AllTrop experiments when the tropospheric wave forcing is constrained; we also note the possibility that the imposed tropospheric wave fields may already contain an element of influence from the upper atmosphere via wave reflection). Nevertheless, compared with the control simulation, the timing of the warmings becomes more realistic. At 10 hPa, the control run has a marked absence of SSW events (negative winds) in January (Supplementary Fig. 6, left column). In both UpStrat-Eq-ERA and UpStrat-Eq-MERRA, there are more events in January, and they are more realistically clustered around the observed SSW date. The timing of these events can be traced upwards to the 1 hPa level, where the control run shows clusters of negative winds in late November/early December and again in February, while in both UpStrat-Eq experiments the November/December cluster is less evident and the January cluster is more prominent. The clustering is further improved if

the full **u**, **v** and $T$ fields are relaxed, and the relaxation is extended to between 0 and 30ºN, particularly in the case of the MERRA experiment.

In summary, relaxing the SAO alone without an accurate simulation of the tropospheric wave forcing does not reproduce the SSW event in the ensemble-mean. This is unsurprising, given the lack of wave forcing constraint from just the SSTs alone. Nevertheless, more ensemble members display warming events clustered around the correct timing than in the control run. Together with the AllTrop experiments, the results confirm that neither constraint is sufficient on its own to adequately reproduce the warming in 2008/9. They highlight a sensitivity to the SAO, a region of the atmosphere that is relatively neglected in seasonal and climate model development.

**Implications for seasonal forecasts**. In reality, we have no specific knowledge of the evolution of either the tropospheric wave forcing or the SAO winds ahead of time. Seasonal forecast skill requires accurate initialisation of the model in early winter and a capability to simulate the relevant physical and thermodynamic processes going forward in time, using an ensemble approach to provide estimates of the likelihood of a particular outcome. In recent years, much attention has focused on improving the capabilities of forecast models to simulate equatorial winds in the lower stratosphere, the region dominated by the QBO. However, capturing processes that determine the SAO is arguably more important. Long radiative timescales in the lower stratosphere mean that correct initialisation of the QBO phase is likely to be reasonably well maintained in the following winter months[44]. In contrast, much shorter radiative timescales in the upper stratosphere/mesosphere mean that memory of the SAO initial conditions is lost within just a few days.

Here, we further explore the potential for improving probabilistic seasonal forecasts of warming events. In experiment UpStrat-EqClim, we relax **u**, **v** and $T$ fields in the upper equatorial stratosphere (0–30ºN, above 5 hPa) towards ERA-Interim climatological fields (Supplementary Fig. 8). The experiment is therefore more closely aligned to a typical seasonal forecast with no information about the particular winter apart from the initial conditions and lower boundary SSTs (which are predictable to some degree in seasonal systems). We note that standard operational seasonal forecast models are more sophisticated than our model set-up, with a higher horizontal resolution, coupled ocean processes with initialised ocean state and they are usually initialised in November or later. Nevertheless, our experiments are a useful indication of the potential for improvement if mean biases in the tropical upper stratosphere can be reduced. UpStrat-EqClim can be directly compared with the control simulation, the only difference being the relaxation towards the climatological SAO. As in the control and UpStrat-Eq experiments, the lack of a strong constraint on the tropospheric circulation means that the UpStrat-EqClim ensemble spread is large (Supplementary Fig. 8), but the timing of the SSWs is once again improved with more ensemble members predicting warming events within 15 days of the observed event (13/50) than in control (2/50).

In a second exploratory experiment, the potential influence of directly correcting the easterly bias in the equatorial mesosphere is examined. In contrast to the reanalyses, MLS and SABER satellite observations show relatively strong westerlies up to 40 ms$^{-1}$ peakings in the mesosphere at ~0.1 hPa that are present throughout the year[40,41]. In the UpStrat-Eq-Clim40 experiment, the ERA-interim climatological **u** fields were modified by adding 40 m/s above 0.5 hPa (see Supplementary Fig. 6). The relaxation was applied only to the zonal winds between 0 and 10ºN, and only in the mesosphere above 0.5 hPa. Relaxing **u** above this higher level

has the added benefit of allowing the model more freedom to determine the onset of the SAO easterly phase itself. The results (Supplementary Fig. 8) are promising in terms of the timing of the clusters of warming events at both 1 hPa and 10 hPa, with a clear suggestion of a warming event in late January, as observed.

## Discussion

Our experiments have highlighted a sensitivity of extreme polar vortex events to the SAO, hitherto unrecognised by the seasonal forecasting community. The results suggest that improved representation of the SAO to correct an easterly bias is likely to improve probabilistic predictions of the timing and depth of sudden stratospheric warming events and hence their impacts on surface weather. Standard approaches such as bias-correcting the flow once the forecast has been performed will not successfully remove the impacts of the bias, since we have shown that the impacts are non-linear, consistent with the known properties of wave mean-flow interaction and sudden warmings. A relatively small SAO anomaly can influence the flow and substantially impact the evolution throughout the winter.

Correcting the easterly bias in a seasonal forecast model by relaxing towards climatological fields or adding a mesospheric westerly forcing throughout the evolution has provided a pragmatic and relatively successful approach in the 2008/9 case study examined here. However, this approach may not be appropriate for other years (see Supplementary Table 1 for corresponding experiments for the other two case studies) since the SAO region shows significant interannual and decadal-scale variability[40,45,46]. The improved model parameterisation of the underlying physical processes that determine the SAO would be preferable, so that the SAO evolution in individual winters can be better represented. There is also much uncertainty in the fidelity of the available reanalysis data in the SAO region, which are known to also underestimate the strength of the westerly flow[40,41] when compared with satellite observations. Improved observational constraints are thus required to validate and improve both seasonal forecast models and the reanalysis datasets, and we suggest this should be a priority for the future.

## Methods

**The model**. Model experiments used the atmosphere-only configuration of the Met Office Unified Model[47] (UM, GA7.0 version 10.3) at N96 resolution (~1.25º latitude, 1.875º longitude) and 85 vertical levels extending to 85 km (0.0053 hPa). A non-orographic gravity wave parametrisation scheme generates a realistic QBO. The 2008/9 monthly averaged SSTs and sea-ice concentrations were prescribed at the lower boundary[48].

**Relaxation scheme**. Selected model fields were relaxed towards the 6-h three-dimensional European Centre for Medium-Range Weather Forecasts (ECMWF) ERA-Interim (ERA-I) or Modern-Era Retrospective analysis for Research and Applications, version 2 (MERRA2) reanalysis fields at each timestep using a Newtonian scheme[49] $\Delta X = G\Delta t$ ($X_{analysis} - X_{model}$) equation where X denotes the field, $\Delta X$ the increment applied over the time interval $\Delta t$ and $G$ is the relaxation parameter (a constant). A 6-h relaxation timescale was employed. The relaxation was applied only above 2.5 km to avoid the atmospheric boundary layer. A linear tapering to zero was applied between 1 and 5 km below the specified height (depending on the height region since the vertical model spacing is not uniform) to avoid abrupt changes at the edge of the relaxation region. In the troposphere, the relaxation was applied to the zonal (**u**) and meridional (**v**) winds and temperatures ($T$). In the stratosphere, the relaxation was applied only to the zonal winds unless otherwise stated. For applied relaxation up to the tropopause, the closest model level to the local lapse-rate tropopause was identified. Where employed, the ERA-Interim climatological fields were derived by averaging the 6-h fields for 1979–2018.

**Experimental design**. Experiments comprised 50-member ensembles generated using a stochastic kinetic energy backscatter scheme that produces perturbed tendencies in the model's primitive equations[50]. Initial conditions were generated by running the model for 2 months from July 1 relaxing the winds and

temperatures everywhere to the three-dimensional ERA-Interim fields. All experiments commenced on September 1.

**Diagnostics**. The number of sudden warming events for Supplementary Table 1 was identified by determining when the zonally averaged zonal winds at 60°N, 10 hPa zonal winds became easterly. Once an SSW was identified, no further events were counted in that year. Statistical significance in Supplementary Table 1 was calculated using Monte Carlo resampling methods. Time series from the control experiment and the appropriate experiment were combined and then randomly split into two dummy 50-member ensembles. By comparing the (absolute) difference in correlation or warming number with 10,000 random distributions from the Monte Carlo resampling the likelihood that the measured difference occurred by chance could be calculated.

## Data availability

The model data that support the findings of this study are available from the corresponding author upon reasonable request. ERA-Interim data are available from https://www.ecmwf.int/en/forecasts/datasets/reanalysis-datasets/era-interim. MERRA2 data are available from https://gmao.gsfc.nasa.gov/reanalysis/MERRA-2/. Sunspot data used to estimate the phase of the solar cycle in Supplementary Table 1 are available from https://www.swpc.noaa.gov/products/solar-cycle-progression.

## Code availability

The Unified Model is available for use outside the Met Office through a licensing agreement. Further information can be found at https://www.metoffice.gov.uk/research/approach/modelling-systems/unified-model/index.

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

## Acknowledgements

We acknowledge the use of the Monsoon computing system, a collaborative facility under the Joint Weather and Climate Research Programme, a strategic partnership between the Met Office and the Natural Environment Research Council (NERC). We also acknowledge the UKCA team who developed the relaxation scheme. H.L. and C.O. were supported by the NERC ACSIS project (Atlantic Climate System Integrated Study). L.J.G. was supported by the NERC ACSIS project and the NERC GOTHAM (Globally Observed Teleconnections in Hierarchies of Atmospheric Models; NE/P006779/1) project. M.J.B. was supported by the Oxford University NERC Doctoral Training Programme. J.K. and M.A. were supported by the Met Office Hadley Centre Climate Programme funded by the UK Government Department of Business, Energy and Industrial Strategy and Department of Environment, Food and Rural Affairs.

## Author contributions

L.J.G. and M.B. designed and performed the research and analysed the data; L.J.G., M.B., J.K., M.A., H.L., C.O. and J.A. wrote the paper.

## Competing interests

The authors declare no competing interests.
