## [Peer Review File · Nature Communications]

Reviewers' comments, first round:

Reviewer #1 (Remarks to the Author):

Forecasting Extreme Stratospheric Polar Vortex Events

by Gray et al

Recommendation: minor revisions

This paper argues that the intra-seasonal timing of SSW could be better predicted if winds in the tropical upper stratosphere were more realistically simulated by models. Taking the 2009 SSW as a case study, relaxing the troposphere does lead to an improved simulation of the SSW as compared to a simulation with just SSTs, but there is no deterministic predictability. Relaxing the upper stratosphere at all latitudes helps with the timing of the SSW, but the SSW does not propagate downwards. Nudging both the troposphere and the tropical upper stratosphere leads to a highly accurate prediction of the vortex evolution. The authors also performed similar simulations in which both the troposphere and tropical upper stratosphere were relaxed for two other SSW, and again the SSW timing was determined deterministically.

The paper is very convincing in demonstrating that the tropical upper stratosphere helps to dictate the seasonal evolution of the vortex, and I have only suggestions on some relevant work the authors appear to have missed as well as a suggestion that I leave for future work.

General Comments on Content:

The authors performed a UpStrat simulation where relaxation is applied at subpolar latitudes as well as tropical latitudes. This overly determines the timing of the SSW (as is evident in Figure 1d). An experiment in which only relaxation is imposed in the tropical upper stratosphere would be much more convincing. It would also allow a more meaningful dissection of the skill in the AllTrop-UpStrat-Eq experiment into a component due to AllTrop, a component due to UpStrat-Eq, and a nonlinear interaction component. Such a dissection would help the reader intuit the relative importance of the troposphere versus the equatorial upper stratosphere, while at present it is hard to draw any such conclusions. I deliberated over whether to include this as a major revision the authors must make otherwise I could not recommend publication, but decided that the manuscript is already convincing of its main points. However, I strongly recommend the authors consider performing this extra analysis for future work.

Minor comments:

p. 3 (first page of introduction), Line 28 It should be noted that some SSWs are predictable for much longer, such as the 2019 SSW (Rao et al 2019)

page 10 (last page of summary), line 8: This statement regarding the maintenance of the QBO can be made stronger. Figure 2 of Garfinkel et al 2018 shows that some subseasonal models (i.e. those that contain an appropriate GW scheme) can indeed maintain a QBO, but not all.

It is worth noting in the introduction and discussion that there has been work on the extent to which the MJO and blocking dictate the timing of SSW within winter. See Garfinkel et al 2012, Kang and Tziperman 2017, and Garfinkel and Schwartz 2017 for the MJO, and Bao et al 2017, Peings 2019, and Attard and Lang 2019 for blocks. Hence this is not the first study to tackle the problem of intra-seasonal timing of SSW, but it certainly provides a new perspective.

The caption for supplemental figure 3 refers to figure S1, but I think this should be referring to Figure S2

Did the authors perform AllTrop or UpStrat experiments for the other two SSWs? If yes, adding them to the relevant figures would be a nice addition.

Attard, H.E. and A.L. Lang, 2019: Troposphere–Stratosphere Coupling Following Tropospheric Blocking and Extratropical Cyclones. *Mon. Wea. Rev.*, 147, 1781–1804, <https://doi.org/10.1175/MWR-D-18-0335.1>

Bao, M., Tan, X., Hartmann, D.L. and Ceppi, P., 2017. Classifying the tropospheric precursor patterns of sudden stratospheric warmings. *Geophysical Research Letters*, 44(15), pp.8011-8016.

Kang, W. and Tziperman, E., 2017. More frequent sudden stratospheric warming events due to enhanced MJO forcing expected in a warmer climate. *Journal of Climate*, 30(21), pp.8727-8743.

Garfinkel, C.I., Schwartz, C., Domeisen, D.I., Son, S.W., Butler, A.H. and White, I.P., 2018. Extratropical atmospheric predictability from the quasi-biennial oscillation in subseasonal forecast models. *Journal of Geophysical Research: Atmospheres*, 123(15), pp.7855-7866.

Garfinkel, C.I. and Schwartz, C., 2017. MJO-related tropical convection anomalies lead to more accurate stratospheric vortex variability in subseasonal forecast models. *Geophysical research letters*, 44(19), pp.10-054.

Garfinkel, C.I., Feldstein, S.B., Waugh, D.W., Yoo, C. and Lee, S., 2012. Observed connection between stratospheric sudden warmings and the Madden-Julian Oscillation. *Geophysical Research Letters*, 39(18).

Peings, Y., 2019. Ural Blocking as a Driver of Early-Winter Stratospheric Warmings. *Geophysical Research Letters*, 46(10), pp.5460-5468.

Rao, J., Garfinkel, C.I., Chen, H. and White, I.P., 2019. The 2019 New Year stratospheric sudden warming and its real-time predictions in multiple S2S models. *Journal of Geophysical Research: Atmospheres*, 124(21), pp.11155-11174.

Reviewer #2 (Remarks to the Author):

Forecasting Extreme Stratospheric Polar Vortex Events
L. J. Gray et al.

The authors present novel numerical experiments designed to identify regions in the atmosphere that are important for the seasonal forecasting of stratospheric sudden warmings. These are highly dynamical events in which the climatological westerly flow in the winter time stratosphere is rapidly reversed; they occur roughly two out of every three winters in the Northern Hemisphere but their timing within the winter is highly variable.

They exert a significant influence on surface weather conditions in the Northern Hemisphere, so that knowledge that such an event either is or will be occurring is of significant value. The current

state of the art in forecasting these events is about two weeks, but the factors that control the timing and onset of the stratospheric variability are debated.

The results presented here indicate that circulation patterns in the tropical upper stratosphere (30 to 50 km altitude, roughly) play an unexpectedly important role in determining the timing of these events. This is an under-observed region of the atmosphere that receives relatively little attention from modeling centers, particularly those concerned with operational forecasting on subseasonal to seasonal timescales. It is unlikely that this region is high on anyone's priority list for model improvements, and so this result, if correct, does have the potential to significantly change priorities in this area. Indeed I would anticipate some resistance to these ideas on the part of the operational centers (or at least to investing computational resources in resolving upper stratospheric processes); this is an argument given in support of publication.

There are two experiments that point toward the importance of the upper stratosphere: the AllTrop-UpStrat-Eq experiment and the UpStrat-EqClim experiment.

In the first case the highlight is that the nudging in the AllTrop-UpStrat-Eq case is sufficient to essentially entirely constrain the extratropical stratospheric flow, given perfect knowledge of the tropospheric evolution. With just the latter, the exact timing of the warming is not deterministically predictable suggesting that there is enough error growth within the stratosphere to disrupt forecasts.

On the face of it this suggests that providing information about the tropical upper stratospheric evolution is sufficient to fully constrain this error growth. My significant concern here is about the technical approach to including this information; this is done by relaxing the full three dimensional flow towards the state of the reanalysis. Because of the tight coupling between temperatures and the wind field associated with the Coriolis effect, the effects of such nudging become strongly non-local in the extratropics. Nudging at 30 N will provide a significant *direct* constraint on both the zonal mean flow and the planetary wave field further poleward and below the 5 hPa level. The methods section also mentions some tapering in latitude and height, it's not clear whether that tapering is done equatorwards or polewards of 30 N. If the former is the case than I would be very concerned that there is a lot of extratropical information being imposed through the nudging. If the latter

is the case I would still be worried. My suspicion is that the additional constraint implied by, say, Figure 2, is not purely tropical in origin, and may be a bit less remarkable than it first appears.

To really show that this error growth arises from the semi-annual oscillation it would be much more convincing if the same effects can be shown with nudging imposed only within, say, 15 degrees of the equator, within the low-latitude regime discussed by Haynes (1998).

A related question I have is to what extent this error growth is chaotic in nature (and could be expected even with a perfect model given imperfect initial conditions), and to what extent it has more to do with model bias. I think the claim is more the latter; if this is the case the good news for operational centers is that simply imposing a correct climatological evolution should lead to improved forecasts.

The second key experiment, UpStrat-EqClim would seem to back that up, and I think it lends key support that the equatorial nudging does not fully constrain the extratropical flow (otherwise relaxing towards climatology would suppress the planetary wave variability in the vortex). But here also: if the model bias is simply being improved, is the improved forecast really just a result of an improved boundary condition for the extratropical waves, or do we really need better initial conditions for the SAO? Of course the former is still important, but would have somewhat weaker implications for investing modeling resources into the tropical upper stratosphere.

I think these are significant concerns regarding the overall implication of the results presented, and I would be eager to hear the authors' response. If the importance of the upper tropical stratosphere can really be established I would think this is highly worthy of publication in Nature Communications.

Finally, an additional result is discussed in the abstract, that imposing the tropospheric evolution is not sufficient to constrain the evolution of the vortex, is certainly interesting and valuable as well, though a very similar result has recently been obtained by de la Camara et al. (2017; full reference given below). I don't see this as the central claim to novelty by the present authors, but a citation should be added.

A few minor comments:

Are the correlations reported in Table S1 based on deseasonalized wind anomalies? I would think DJF correlations might be more relevant to isolating the dynamical variability of interest. Have similar UpStrat-EqClim experiments been performed for other years? If so how do these corresponding correlations look in other cases?

p4 l26 to 28 needs a bit of editing.

References

de la Camara et al. (2017). "Sensitivity of sudden stratospheric warmings to previous stratospheric conditions." *J. Atmos. Sci.* 74: 2857–2877. doi: 10.1175/JAS-D-17-0136.1.

P.H. Haynes (1998) The latitudinal structure of the quasi-biennial oscillation Q. *J. R. Meteorol. Soc.* 124: 2645-2670 doi: 10.1002/qj.49712455206

Noguchi et al. (2019) doi: 10.1002/essoar.10501209.1

Dear Editor and Reviewers.

Thank you for your thoughtful comments and the time you have spent reading and reviewing our paper. It is very much appreciated. We have substantially revised the paper, in light of the comments. The major changes are summarised below, followed by a detailed response to each of the reviewers.

Summary of major changes made to the paper:

1. *All experiments have been repeated with stratospheric relaxation applied only to the zonal wind fields between 0-10N instead of 0-30N, in response to comments from reviewer 2. This makes virtually no difference to the AllTrop-UpStrat-Eq results, confirming that the imposed relaxation does not extend sufficiently far northward to directly influence the high-latitude fields.*
2. *Additional experiments have been performed with relaxation only in the equatorial upper stratosphere / lower mesosphere, in response to comments from reviewer 1. These have been performed using both ERA Interim and MERRA2, to investigate the impact of differences between reanalysis fields in the SAO region. Not surprisingly, given the lack of constraint on tropospheric forcing, the ensemble spread is large but the timing of the warmings is nevertheless slightly improved.*
3. *The major experiments (Control, AllTrop and AllTrop-UpStrat-Eq, UpStrat-Eq-Clim) have now been performed for all three case study years and the results are summarised in the supporting figures and Table S1.*
4. *Figure 4 (which previously served to highlight data on the distribution of SSWs that are already provided in Table S1) has been replaced by a figure to show the differences in equatorial winds from the 2008/9 AllTrop-UpStrat-Eq and AllTrop experiments.*
5. *Further discussion of the differences between reanalysis fields and satellite observations of the SAO region has been added, and an additional experiment performed to examine the impact of directly correcting the easterly bias in the lower mesosphere by adding a westerly component to the zonal wind field above the 0.5 hPa level.*

Responses to the individual review comments:

Reviewer #1 (Remarks to the Author):

Forecasting Extreme Stratospheric Polar Vortex Events by Gray et al
Recommendation: minor revisions

This paper argues that the intra-seasonal timing of SSW could be better predicted if winds in the tropical upper stratosphere were more realistically simulated by models. Taking the 2009 SSW as a case study, relaxing the troposphere does lead to an improved simulation of the SSW as compared to a simulation with just SSTs, but there is no deterministic predictability. Relaxing the upper stratosphere at all latitudes helps with the timing of the SSW, but the SSW does not propagate downwards. Nudging both the troposphere and the tropical upper stratosphere leads to a highly accurate prediction of the vortex evolution. The authors also performed similar simulations in which both the troposphere and tropical upper stratosphere were relaxed for two other SSW, and again the SSW timing was determined deterministically.

The paper is very convincing in demonstrating that the tropical upper stratosphere helps to dictate the seasonal evolution of the vortex,

We thank the reviewer for this very positive comment

and I have only suggestions on some relevant work the authors appear to have missed as well as a suggestion that I leave for future work.

Thank you. We have added your suggested citations (see below).

General Comments on Content:

The authors performed a UpStrat simulation where relaxation is applied at subpolar latitudes as well as tropical latitudes. This overly determines the timing of the SSW (as is evident in Figure 1d). An experiment in which only relaxation is imposed in the tropical upper stratosphere would be much more convincing.

We have performed this additional experiment, as suggested, only relaxing the zonal winds in the upper equatorial stratosphere above ~5hPa 0-10N. In fact we carried out two different versions, towards different reanalysis datasets (ERA-Interim and MERRA2) since this region of the atmosphere is not well constrained by observations and there is much variation between the different reanalyses. The results of the experiments confirm that neither the troposphere nor equatorial upper stratosphere on its own is sufficient to correctly simulate the timing of the SSWs. The results are described in the section 'The Role of the Equatorial Upper Stratosphere' on page 7.

It would also allow a more meaningful dissection of the skill in the AllTrop-UpStrat-Eq experiment into a component due to AllTrop, a component due to UpStrat-Eq, and a nonlinear interaction component. Such a dissection would help the reader intuit the relative importance of the troposphere versus the equatorial upper stratosphere, while at present it is hard to draw any such conclusions.

Yes, agreed, although dissection of the skill is not straightforward and we have avoided discussion of the relative contributions. Although we choose to separate the forcing into just these two contributions for simplicity (AllTrop, UpStrat_Eq), in reality there are many different processes that contribute to the tropospheric forcing, and assigning relative importance may not be entirely appropriate. Few would doubt the importance of simulating the tropospheric wave forcing and it is not our intention to suggest that improving the equatorial upper stratosphere / mesosphere is more important (or even equally important). The main message is that good simulation of both regions is required. It may be appropriate, and certainly of interest to seasonal forecasters, to compare the relative merits, say, of investment to improve specific details of the tropospheric flow (perhaps how well blocking is simulated) compared to improving details of the equatorial mesospheric flow, but this would need a separate study.

I deliberated over whether to include this as a major revision the authors must make otherwise I could not recommend publication, but decided that the manuscript is already convincing of its main points. However, I strongly recommend the authors consider performing this extra analysis for future work.

Thank you. As you will see we have taken up your suggestion and hope that you agree the results have extended the study in a useful direction.

Minor comments:

p. 3 (first page of introduction), Line 28 It should be noted that some SSWs are predictable for much longer, such as the 2019 SSW (Rao et al 2019) *Text amended and ref added*

page 10 (last page of summary), line 8: This statement regarding the maintenance of the QBO can be made stronger. Figure 2 of Garfinkel et al 2018 shows that some subseasonal models (i.e. those that contain an appropriate GW scheme) can indeed maintain a QBO, but not all. *Reference to Garfinkel et al 2018 added*

It is worth noting in the introduction and discussion that there has been work on the extent to which the MJO and blocking dictate the timing of SSW within winter. See Garfinkel et al 2012, Kang and Tziperman 2017, and Garfinkel and Schwartz 2017 for the MJO, and Bao et al 2017, Peings 2019, and Attard and Lang 2019 for blocks. Hence this is not the first study to tackle the problem of intra-seasonal timing of SSW, but it certainly provides a new perspective.

All suggested refs added

The caption for supplemental figure 3 refers to figure S1, but I think this should be referring to Figure S2

Corrected – thank you.

Did the authors perform AllTrop or UpStrat experiments for the other two SSWs? If yes, adding them to the relevant figures would be a nice addition.

Done – see new figures S3 and S4.

Reviewer #2 (Remarks to the Author):

Forecasting Extreme Stratospheric Polar Vortex Events

L. J. Gray et al.

The authors present novel numerical experiments designed to identify regions in the atmosphere that are important for the seasonal forecasting of stratospheric sudden warmings. These are highly dynamical events in which the climatological westerly flow in the winter time stratosphere is rapidly reversed; they occur roughly two out of every three winters in the Northern Hemisphere but their timing within the winter is highly variable.

They exert a significant influence on surface weather conditions in the Northern Hemisphere, so that knowledge that such an event either is or will be occurring is of significant value. The current state of the art in forecasting these events is about two weeks, but the factors that control the timing and onset of the stratospheric variability are debated.

The results presented here indicate that circulation patterns in the tropical upper stratosphere (30 to 50 km altitude, roughly) play an unexpectedly important role in determining the timing of these events. This is an under-observed region of the atmosphere that receives relatively little attention from modeling centers, particularly those concerned with operational

forecasting on subseasonal to seasonal timescales. It is unlikely that this region is high on anyone's priority list for model improvements, and so this result, if correct, does have the potential to significantly change priorities in this area.

Thank you for noting this – we have performed a number of extra experiments to back up the result that shows a significant influence from this region (see details provided below).

Indeed I would anticipate some resistance to these ideas on the part of the operational centers (or at least to investing computational resources in resolving upper stratospheric processes); this is an argument given in support of publication.

Yes, agreed

There are two experiments that point toward the importance of the upper stratosphere: the AllTrop-UpStrat-Eq experiment and the UpStrat-EqClim experiment.

In the first case the highlight is that the nudging in the AllTrop-UpStrat-Eq case is sufficient to essentially entirely constrain the extratropical stratospheric flow, given perfect knowledge of the tropospheric evolution. With just the latter, the exact timing of the warming is not deterministically predictable suggesting that there is enough error growth within the stratosphere to disrupt forecasts.

On the face of it this suggests that providing information about the tropical upper stratospheric evolution is sufficient to fully constrain this error growth. My significant concern here is about the technical approach to including this information; this is done by relaxing the full three dimensional flow towards the state of the reanalysis. Because of the tight coupling between temperatures and the wind field associated with the Coriolis effect, the effects of such nudging become strongly non-local in the extratropics. Nudging at 30 N will provide a significant *direct* constraint on both the zonal mean flow and the planetary wave field further poleward and below the 5 hPa level. The methods section also mentions some tapering in latitude and height, it's not clear whether that tapering is done equatorwards or polewards of 30 N.

If the former is the case than I would be very concerned that there is a lot of extratropical information being imposed through the nudging. If the latter is the case I would still be worried. My suspicion is that the additional constraint implied by, say, Figure 2, is not purely tropical in origin, and may be a bit less remarkable than it first appears. To really show that this error growth arises from the semi-annual oscillation it would be much more convincing if the same effects can be shown with nudging imposed only within, say, 15 degrees of the equator, within the low-latitude regime discussed by Haynes (1998).

We agree that this is a concern and have repeated all the experiments, relaxing only in the region 0-10N with no latitudinal tapering (experiments showed that the tapering was not required for the stability of the model and made no difference to the results, so we left it out to avoid confusion). We also amended the relaxation so that it only relaxed the zonal winds and not the meridional winds or temperatures, which is much closer in concept to a parameterisation of the wave forcing. In this way we more closely represent the action of momentum transfer from waves (most likely Kelvin waves or gravity waves) to the mean flow and allow the meridional winds and temperatures to adjust accordingly. We kept the vertical tapering in place, so where we relax above 5 hPa the relaxation extends slightly below this level but certainly not as far down as 10 hPa.

As a result of these changes there is some increased ensemble spread in the SSW evolution (see revised figure 2) but the differences are barely discernable and the essential timing of the SSW is still extremely well simulated. We also repeated this experiment for the two additional examples in 1988/9 and 2005/6 with the same result (new figures S3 and S4).

The Methods section has been revised to reflect these changes, and all sections of the text / figure captions that refer to relaxation between 0-30N have been removed and replaced by 0-10N.

A related question I have is to what extent this error growth is chaotic in nature (and could be expected even with a perfect model given imperfect initial conditions), and to what extent it has more to do with model bias. I think the claim is more the latter; if this is the case the good news for operational centers is that simply imposing a correct climatological evolution should lead to improved forecasts.

The second key experiment, UpStrat-EqClim would seem to back that up, and I think it lends key support that the equatorial nudging does not fully constrain the extratropical flow (otherwise relaxing towards climatology would suppress the planetary wave variability in the vortex).

Yes, repeating our experiments to relax only to u between 0-10N has confirmed that the equatorial nudging does not fully constrain the extra-tropical flow.

The experiments suggests that the problem lies primarily with the easterly bias. We have done some more work to try to pin this down further – see the new text in section ‘Implications for seasonal forecasts’ on page 10, including an additional experiment. Relaxing towards climatological SAO fields seems a reasonable first approach but see our caveats noted in the next response.

But here also: if the model bias is simply being improved, is the improved forecast really just a result of an improved boundary condition for the extratropical waves,

Our results suggest that improved boundary conditions for the extratropical waves is the key (c.f. where we relax only to the U fields between 0-10N). But, as mentioned above, it’s not conclusive that simply relaxing to climatology in the upper equatorial stratosphere will necessarily improve seasonal forecasts – a separate study to explore this would be very interesting, and hopefully the seasonal forecast centres might be motivated by this paper to do so. One would hope that correcting such a large bias would lead to an improvement, particularly in years where the SAO evolution resembles the climatology (e.g. in solar cycle neutral years). But (while not wishing to be too pessimistic) there are many reasons why this might not result in a positive impact, including uncertainties in the observed SAO climatology and correcting a bias in one region can often reveal other issues. We have added some more text on the uncertainties in the observed SAO climatology (in the ‘Summary and Discussion’ section, page 11).

or do we really need better initial conditions for the SAO?

The radiative timescale in the upper stratosphere is very short, so initial conditions are not the issue here. The question is whether the precise evolution of the SAO in a particular year

needs to be modelled accurately. This will need further study, as discussed above. A further interesting study (if relaxing to climatology is found to be insufficient in many other years) would be to relax to an adapted SAO climatology that takes into account the phase of the QBO / solar cycle.

Of course the former is still important, but would have somewhat weaker implications for investing modeling resources into the tropical upper stratosphere.

Yes, if further studies find that the precise evolution of the SAO is important then additional resources would be needed not only in terms of modelling but also in observations of this region.

I think these are significant concerns regarding the overall implication of the results presented, and I would be eager to hear the authors' response. If the importance of the upper tropical stratosphere can really be established I would think this is highly worthy of publication in Nature Communications.

We hope the reviewer will agree that, with the revisions and additional experiments, this paper is indeed worthy of publication in Nature Comms, to highlight the importance of the upper tropical stratosphere and serve to trigger further studies of this neglected region.

Finally, an additional result is discussed in the abstract, that imposing the tropospheric evolution is not sufficient to constrain the evolution of the vortex, is certainly interesting and valuable as well, though a very similar result has recently been obtained by de la Camara et al. (2017; full reference given below). I don't see this as the central claim to novelty by the present authors, but a citation should be added.

Wording of the abstract has been amended, and reference added (page 5).

A few minor comments:

Are the correlations reported in Table S1 based on deseasonalized wind anomalies? I would think DJF correlations might be more relevant to isolating the dynamical variability of interest.

No, they are based on the actual winds. We have replaced the correlations by DJF correlations instead of Sept-April correlations, as suggested.

Have similar UpStrat-EqClim experiments been performed for other years? If so how do these corresponding correlations look in other cases?

Yes – they are now included in Table S1.

p4 l26 to 28 needs a bit of editing. *Done*

Reviewer Comments, second round

Reviewer #1 (Remarks to the Author):

The authors satisfactorily addressed my comments. Here are just a few more very minor suggestions before acceptance.

page 3, line 31: a recent paper which discusses the potential for probabilistic forecasting is Domeisen et al 2020

Domeisen, Daniela IV, Amy H. Butler, Andrew J. Charlton-Perez, Blanca Ayarzagüena, Mark P. Baldwin, Etienne Dunn-Sigouin, Jason C. Furtado et al. "The role of the stratosphere in subseasonal to seasonal prediction: 2. Predictability arising from stratosphere-troposphere coupling." *Journal of Geophysical Research: Atmospheres* 125, no. 2 (2020): e2019JD030923.

page 11 line 10 : seasonal forecast*ing* community

Reviewer #2 (Remarks to the Author):

I would like to thank the authors for their careful and considered response to my comments on the first version of the

ir manuscript. I am satisfied that the extratropical influence of nudging out to 30 degrees from the equator is not essential for obtaining an improved representation of polar variability within the stratosphere, and that the SAO winds are truly an important boundary condition. I also appreciate the additional discussion of the role of the SAO anomalies from a seasonal prediction point of view - I hope (and expect) that this paper will serve to encourage further efforts to model and understand this region of the tropical atmosphere.

My only additional question is to what extent the authors feel that mechanisms suggested by Kodera and Kuroda (2002) (doi:10.1029/2002JD002224) is relevant to the influence identified here. This work (along with a series of related studies) also implied the importance of the tropical upper stratosphere from a diagnostic perspective.

I apologize for the delay in this second review. In view of the fact that I am generally convinced by the arguments put forth I do not wish to recommend any further revisions but instead will urge its publication as is.

Point by point response to reviewer comments:

REVIEWERS' COMMENTS:

Reviewer #1 (Remarks to the Author):

The authors satisfactorily addressed my comments. Here are just a few more very minor suggestions before acceptance.

page 3, line 31: a recent paper which discusses the potential for probabilistic forecasting is Domeisen et al 2020

Domeisen, Daniela IV, Amy H. Butler, Andrew J. Charlton-Perez, Blanca Ayarzagüena, Mark P. Baldwin, Etienne Dunn-Sigouin, Jason C. Furtado et al. "The role of the stratosphere in subseasonal to seasonal prediction: 2. Predictability arising from stratosphere-troposphere coupling." *Journal of Geophysical Research: Atmospheres* 125, no. 2 (2020):e2019JD030923.

Reference has been added.

page 11 line 10 : seasonal forecast*ing* community
Corrected

Reviewer #2 (Remarks to the Author):

I would like to thank the authors for their careful and considered response to my comments on the first version of the manuscript. I am satisfied that the extratropical influence of nudging out to 30 degrees from the equator is not essential for obtaining an improved representation of polar variability within the stratosphere, and that the SAO winds are truly an important boundary condition. I also appreciate the additional discussion of the role of the SAO anomalies from a seasonal prediction point of view - I hope (and expect) that this paper will serve to encourage further efforts to model and understand this region of the tropical atmosphere.

My only additional question is to what extent the authors feel that mechanisms suggested by Kodera and Kuroda (2002) (doi:10.1029/2002JD002224) is relevant to the influence identified here. This work (along with a series of related studies) also implied the importance of the tropical upper stratosphere from a diagnostic perspective.

I believe the Kodera and Kuroda paper is certainly pertinent, since the subtropical jet is substantially influenced by the semi annual oscillation. Their approach is from a different, but complementary, viewpoint. I have therefore added a reference to it (now ref 46, referred to on line 29 of page 11).

I apologize for the delay in this second review. In view of the fact that I am generally convinced by the arguments put forth I do not wish to recommend any further revisions but instead will urge its publication as is.